# The Robustness of Detecting Known and Unknown DDoS Saturation Attacks in SDN via the Integration of Supervised and Semi-Supervised Classifiers

Samer Khamaiseh [1,*,†], Abdullah Al-Alaj [2,†], Mohammad Adnan [3] and Hakam W. Alomari [1]

1 Department of Computer Science and Software Engineering, Miami University, Oxford, OH 45056, USA; alomarhw@miamioh.edu
2 Department of Computer Science, Virginia Wesleyan University, Virginia Beach, VA 23455, USA; aalalaj@vwu.edu
3 Department of Computer Information Systems, Yarmouk University, Irbid 21163, Jordan; 2018902090@ses.yu.edu.jo
* Correspondence: khamaisy@miamioh.edu
† These authors contributed equally to this work.

**Abstract:** The design of existing machine-learning-based DoS detection systems in software-defined networking (SDN) suffers from two major problems. First, the proper time window for conducting network traffic analysis is unknown and has proven challenging to determine. Second, it is unable to detect unknown types of DoS saturation attacks. An unknown saturation attack is an attack that is not represented in the training data. In this paper, we evaluate three supervised classifiers for detecting a family of DDoS flooding attacks (UDP, TCP-SYN, IP-Spoofing, TCP-SARFU, and ICMP) and their combinations using different time windows. This work represents an extension of the runner-up best-paper award entitled 'Detecting Saturation Attacks in SDN via Machine Learning' published in the 2019 4th International Conference on Computing, Communications and Security (ICCCS). The results in this paper show that the trained supervised models fail in detecting unknown saturation attacks, and their overall detection performance decreases when the time window of the network traffic increases. Moreover, we investigate the performance of four semi-supervised classifiers in detecting unknown flooding attacks. The results indicate that semi-supervised classifiers outperform the supervised classifiers in the detection of unknown flooding attacks. Furthermore, to further increase the possibility of detecting the known and unknown flooding attacks, we propose an enhanced hybrid approach that combines two supervised and semi-supervised classifiers. The results demonstrate that the hybrid approach has outperformed individually supervised or semi-supervised classifiers in detecting the known and unknown flooding DoS attacks in SDN.

**Keywords:** machine learning; software-defined networking; OpenFlow; DoS saturation attacks





## 1. Introduction

Software-defined networking (SDN) is a new network architecture that allows an intelligent program, known as the controller, to centrally orchestrate the entire network behavior by detaching the control logic from the underlying network infrastructure. The control plane and the data plane in different devices are connected through a communication channel in the form of an application programming interface known as southbound API. OpenFlow saw the first and the most widespread adoption as a de facto standard southbound API for SDNs. Nevertheless, SDNs suffers from various security threats. Kreutz et al. [1] presented various security problems associated with SDN, including application authorization and the potential flooding of both control and data planes. One specific issue is dealing with DoS saturation attacks [2] that may saturate both the data and the control planes.

The DoS attack is one of the most widely used malicious cyberattacks, and its detection is at the forefront of cybersecurity researchers' interest. Finding new ways to protect the SDNs against DoS attacks is a hot topic of research since many large technology companies are attracted to the benefits of SDN technology. Recently, different research works presented different methods focused on detecting TCP-SYN flooding such as SLICOTS [3], AVANT-GUARD [4], and LineSwitch [5]. FloodGuard [6] and FloodDefender [7] use Packet-In rates to detect flooding attacks. However, this may lead to increasing the false positives since a high volume of new normal traffic may increase the rate of Packet-In messages that may exceed a pre-defined threshold [2]. FDAM [8] utilizes the support vector machine (SVM) to detect DoS flooding attacks and the white-list technique to mitigate the flooding attacks. SA-Detector [2] studies the similarity between the normal and attack OpenFlow traffic to detect several types of saturation attacks. Our recent works [9–11] demonstrated that supervised classifiers are useful for detecting the occurrences of these types of attacks.

One way to provide the protection against DoS attacks in SDN is the adoption of machine learning approaches [12–14]. A successful ML-based way to detect DoS attacks in SDN should (1) address the detection of known as well as the unknown saturation attacks, (2) have high classification rate of the attack traffic, (3) have a high accuracy rate, and (4) be able to detect a family of saturation attacks rather than a single attack traffic.

However, from a review of the literature of machine learning DoS flooding detection methods, two main shortcomings can be noted. The first is related to the time window for collecting and analyzing the SDN network traffic, and the second is related to the detection of saturation attacks that are yet unknown [8].

When the time window set for collecting and analyzing the SDN network traffic is long, the detection time becomes relatively long and sufficient for the attacker to flood the network. On the other hand, having a short time window renders the captured traffic amount insufficient for providing reliable and accurate detection results. However, works in the literature, such as [7,14,15], use an arbitrary pre-defined and fixed time window for collecting and analyzing the SDN network traffic. This has a significant impact on the performance of the ML-detection methods as mentioned earlier.

The detection of unknown saturation attacks is a key concern in SDN, which, if not detected in early stages, can render the SDN environment vulnerable to flooding attacks. Many research works show different attack techniques [9–11]. However, they use a single machine-learning classifier to detect limited types and pre-defined (known) saturation attacks (e.g., SYN flooding attack). It is unclear whether they can detect different types of (unknown) saturation attacks.

In this paper, we address the aforementioned issues by providing answers to the following questions:

1. Is it possible to find the proper time window for performing OpenFlow traffic analysis, and if so, what is it?
2. How is the detection performance of ML classifiers affected by the variation of the time window?
3. How effective are the existing ML classifiers for detecting unknown saturation attacks?

To address questions 1 and 2 and discover the proper time window for detecting DoS saturation attacks, we investigate the detection robustness of the widely used supervised classifiers such as K-nearest neighbor (K-NN), support vector machine (SVM), and naive Bayes (NB) using different time window configurations of OpenFlow traffic analysis. In addition, we evaluate the impact of various time windows of traffic analysis on the performance of detecting these attacks by conducting the false-negative analysis.

To address question 3, we evaluate supervised (SVM, K-NN, and NB) and semi-supervised (Isolation Forest, Basic Autoencoder, One-Class SVM, and Variational Autoencoder) classifiers for detecting unknown saturation attacks. The supervised classifiers are evaluated using two-evaluation scenarios: the best-case training method and the worst-case training method. The experimental results of the best-case scenario training method show that the SVM, NB, and K-NN classifiers can detect unknown saturation attacks. How-

ever, the experimental results of the worst-case scenario training method show that the supervised classifiers fail to detect unknown attacks. Thus, we adopt the semi-supervised classification algorithms. The results show that semi-supervised classifiers can detect unknown attacks. Specifically, the variational autoencoder provides a higher detection performance.

To further improve the detection of unknown and known flooding attacks, we proposed the hybrid approach. This approach focuses on taking advantage of two classification algorithms: a supervised algorithm to effectively detect known attacks and semi-supervised algorithm to detect unknown flooding attacks. The results show that the hybrid approach provides higher detection performance for both known and unknown attacks than a single supervised and semi-supervised classifier.

To summarize, the contributions of this paper are as follows:

- To our knowledge, this work is one of the very first of its kind to study the effectiveness of using the supervised and semi-supervised classifiers in detecting the unknown DoS saturation attacks in SDN.
- We propose a hybrid approach that combines the semi-supervised and supervised classifiers to detect the unknown and known DoS flooding attacks.
- Two datasets of different types of saturation attacks are created using the simulated and physical SDN environments which will help in promoting further research in SDN security (we will disclose these datasets and the code after the acceptance of this paper).

The remainder of this paper is organized as follows. Section 2 reviews the related work. Section 3 describes a brief introduction about saturation attacks in SDN. In Section 4, the methodology and the process of collecting OpenFlow traffic and data are given. Section 5 presents the experimental results and discussions. Section 6 describes threat of validity. Finally, Section 7 concludes the paper.

## 2. Related Work

As a new network paradigm that provides agility and programmability, SDN attracts research in industry and academia worldwide. Long et al. [16] proposed a hybrid approach to detect DDoS attacks in SDN using two main modules: (1) the initial detection module that uses the information entropy to detect attack traffic and (2) the detection module that uses the stacked sparse autoencoder and support vector machine classifiers to validate the attack traffic. The proposed hybrid approach may not be able to detect combined multiple unknown DDoS attacks. Polate et al. [17] proposed a new approach to detect and protect the SDN-based supervisory control and data acquisition (SCADA) systems against DDoS attacks. The proposed approach uses two parallel neural networks including long short-term memory (LSTM) and gated recurrent units (GRU) to detect the DDoS attack. They also use SVM to classify the extracted features from the network traffic. This approach has two limitations: First, it may cause computational overhead when it applies to a real-life environment since it uses three machine learning classifiers to detect malicious traffic and classify traffic features. Second, the detection time required by three classifiers will be long, which may enable the attack traffic to overwhelm the entire SDN environment.

Ashraf et al. [12] discussed the possibility of adopting machine learning approaches in SDN to detect DDoS attacks. However, this work did not go so far as to investigate potential approaches that can be used to detect OpenFlow switches targeted by known and unknown saturation attacks. Niyaz et al. [13] presented a multi-vector DoS attacks using the stacked autoencoder (SAE). It processes all received packets for flow computation and attack detection, which requires extensive computational resources, instead of flow sampling. Aizuddin et al. [14] introduced a detection system for DDoS attacks by adopting the Dirichlet process mixture model. With this system, the misclassification rate of the attack traffic is around 50%. Hu et al. [8] introduced the FDAM system to detect and mitigate ICMP, SYN, and UDP flooding attacks. The FDAM detection method uses the SVM classifier to detect these attacks and a white-list approach to block the malicious traffic.

The main shortcoming of the FADM detection method is using the SVM classifier, which cannot detect unknown saturation attacks and requires a long training and prediction time.

Tang et al. [18] introduced a DoS detection system by using the deep neural network (DNN). The accuracy of the proposed detection model is 88.04%, which is relatively low. The proposed system was trained and tested by the NSL-KDD dataset, which is generated by traditional network traffic. Abubakar and Pranggono [15] detected the attack flows by using a neural network. They used the NSL-KDD to train and test the proposed system. Braga et al. [19] adopted the self-organized map (SOM) to develop a lightweight detection system for DDoS flooding attacks against SDNs. The proposed system collects all OpenFlow switches flow entries to extract the SOM classifier features to detect DDoS attacks. Thus, the features extraction process requires a long processing time which may enable the attacker to flood the network. Ye et al. [20] used a trained SVM classifier to detect SYN, ICMP, and UDP attacks. The proposed system includes (1) flow state collection method that is responsible for collecting the flow-table states of the OpenFlow switches, (2) characteristic values extraction to extract the classifier features, and (3) classifier judgment that uses the SVM classifier to detect the attacks. This work represents a simple method for detecting some flooding attacks in SDNs. Ambrosin et al. [5] proposed LineSwitch, which is an OpenFlow module that can be deployed to OpenFlow switches to countermeasure SYN flooding attacks based on probability and blacklisting. This approach focuses only on protecting the SDN environment against the SYN flooding attack. However, our approach provides the capability of detecting a family of saturation attacks, including the SYN flooding attack.

Aziz and Okamura [21] proposed the FlowIDS framework to detect simple mail transfer protocol (SMTP) flooding attacks. They utilized the decision tree (DT) and deep learning (DL) algorithms to detect malicious SMTP traffic. After being trained to identify benign SMTP traffic, the deep learning algorithm and the decision tree classifier were used in detecting any traffic used to launch the flooding attack. Silva et al. [22] introduced the ATLANTIC framework to detect DDoS attacks against the SDN environment. The proposed system uses entropy analysis to detect the deviations of the SDN network traffic to discover the suspicious traffic flows. It also uses a K-means unsupervised model to cluster similar traffic flows. Therefore, an SVM classifier is employed to label flows as malicious or normal. The detection performance of the proposed system is relatively low; it obtained 88.7% accuracy and 82.3% precision, which prevents such systems from being adopted in real life.

Liu et al. [23] proposed the FL-Guard detection and defense system against DDoS attacks in the SDN environment. FL-Guard uses the SVM classifier to detect saturation attacks. However, FL-Guard cannot detect major saturation attacks, such as UDP flooding attacks or unknown saturation attacks. Chen et al. [24] proposed a detection method for domain name service (DNS) and network time protocol (NTP) reflection amplification attacks. This work aims to detect two attacks by using an SVM classifier.

In a prior work [9–11], we focused on different challenges of building machine-learning-based saturation attacks detection methods and presented solutions to these challenges, such as identifying the proper time window for OpenFlow traffic analysis. Most of the related work uses periodic detection methods, which means that the detection method starts with a pre-determined time window, without providing a justification for this time window. We also investigated the K-NN, NB, and SVM classifiers for detecting the saturation attack in both physical and simulated SDN setups. The results show that the detection performance of these classifiers are heavily impacted by the time window of OpenFlow traffic.

In this paper, we extend our work by investigating and evaluating the robustness of supervised and semi-supervised classifiers in detecting the unknown DoS saturation attacks in SDN networks. We also propose an enhanced detection method that can detect both known and unknown saturation attacks in SDN by combining supervised and semi-supervised classifiers.

## 3. SDN Saturation Attacks Overview

In SDNs, a table miss occurs upon the arrival of a new packet which the data plane does not know how to handle, i.e., there is no matching flow entry installed in the flow table of the switch. As a result, a Packet-In message is generated. Depending on the switch memory status, this Packet-In may comprise the header of the received packet when the switch memory is not full, or the whole packet is encapsulated in the Packet-In message and passed to the controller if the switch memory is full. After receiving the Packet-In message, the controller decides the fate of the table-miss packet by sending Packet-Out and Flow-Mod messages to install flow rule(s) in the flow table of the OpenFlow switch. This reactive packet processing exposes a security vulnerability. An attacker may exploit table miss to exhaust the control and data planes' computational resources (i.e., CPU and RAM) and flood the OpenFlow channel by launching a data-to-control plane attack. The saturation effect of this attack is caused by generating and sending vast amounts of Packet-In messages to the SDN controller which drastically consumes the controller's computational resources. Figure 1 shows the CPU utilization of the SDN controller under the UDP flooding attack. Before the attack started, the CPU utilization of the control plane ranges from (20–40%). When the UDP flooding attacks are triggered, the CPU utilization reached up to 90% since the controller is overwhelmed by processing the malicious Packet-In packets.

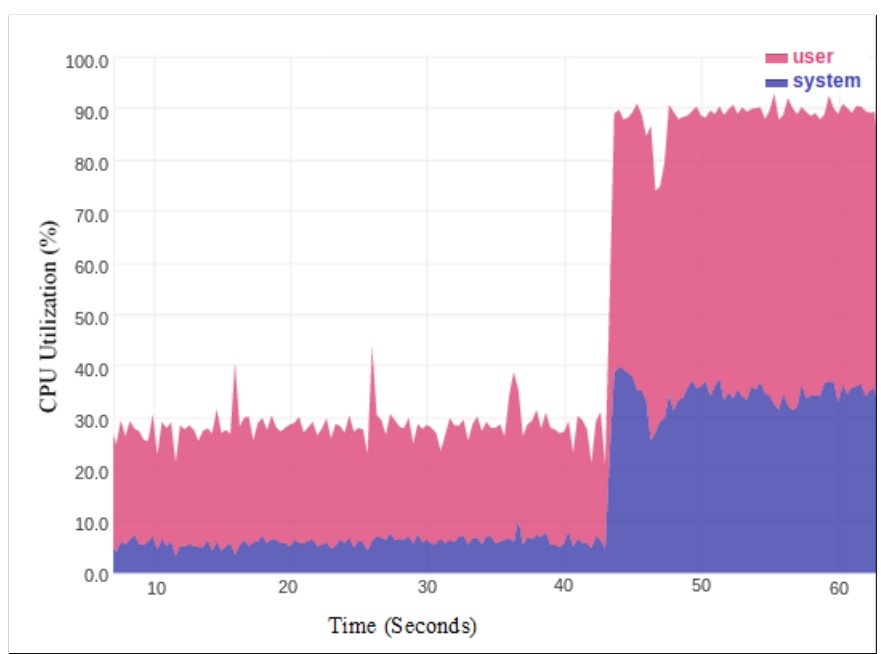

**Figure 1.** Control plane CPU utilization under UDP flooding attack.

Upon a successful data-to-control plane flooding attack, a large number of Packet-Out and Flow-Mod messages are sent by the controller, which leads to a control-to-data plane attack. Therefore, the targeted switch(s) flow tables are filled with fake flow rules, which prevents the benign flow rules from being installed. At this point, the buffer of the victim switch is consumed, and unable to accommodate legitimate new packets. This also exhausts the OpenFlow channel bandwidth, which disables the delivery of OpenFlow messages between the controller and the OpenFlow switches. Figure 2 depicts the impact of the UDP saturation attack on the OpenFlow connection channel bandwidth. Before launching the attack, the average bandwidth of the OpenFlow connection channel is equal to 3.5 Gbps. When the UDP attack targets the SDN environment, the bandwidth of the OpenFlow connection channel drops to 0 Gbps due to the vast amount of OpenFlow messages (e.g., Packet-In and Packet-Out) that are generated by the malicious table miss packets.

To conduct saturation attacks in the SDN network, the attacker can employ the TCP-SYN, UDP, ICMP, IP-Spoofing, and TCP-SARFU flooding attacks, or their combinations

(i.e., hybrid saturation attacks) to generate a large number of Packet-In messages. To defend SDNs against these attacks, it is highly important to detect both known and unknown saturation attacks.

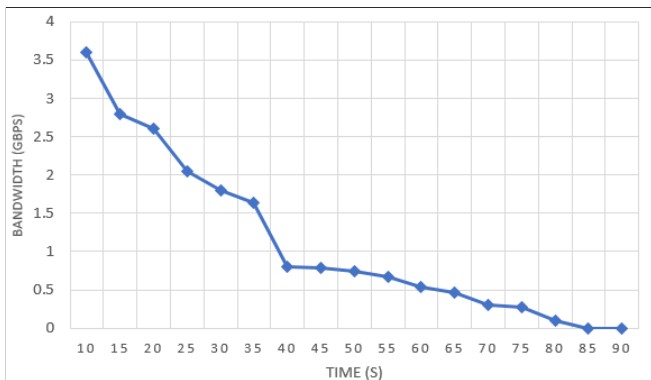

**Figure 2.** The utilization of OpenFlow channel bandwidth under UDP attack.

## 4. Methodology

In this section, we present the process of collecting OpenFlow traffic and data. Then we introduce the feature extraction of the OpenFlow traffic data for different time windows. We briefly introduce the supervised and semi-supervised algorithms used in this paper and our hybrid approach.

### 4.1. The Benign and Malicious OpenFlow Traffic Collection

The OpenFlow traffic is collected by utilizing simulated and physical SDN environments. The physical environment comprises five hosts, one PICA8 switch as an OpenFlow switch, one Floodlight 1.2v Master as an SDN controller, and OpenFlow protocol v1.3. The adopted physical environment is limited by the network topology and scale. Therefore, different SDN simulated environments were created using the Mininet tool. By using the simulated environment, we created different network topologies and scales. Table 1 presents the simulation SDN network configurations, which were obtained from the Mininet examples.

**Table 1.** SDN simulation environment configurations and specifications.

| Parameter | Description | Default Value |
|---|---|---|
| $N^c$ | Number of SDN Controllers | 1 |
| $N^s$ | Number of OpenFlow Switches | 10–200 |
| $N^h$ | Number of Connected Hosts | 50–300 |
| $N^t$ | Network Topology | Star, Ring, Tree, Mesh |

The normal OpenFlow traffic was collected using both simulated and physical SDN environments by utilizing various traffic generation tool that can mimic the load of real-life network traffic and behavior. First, the Cisco TRex realistic traffic generator was employed to mimic the internet traffic via generating concurrent stateless and stateful traffic. It provides the capability to generate L4–L7 traffic via smart replay traffic templates with different rate up to 200 gbps. Second, to generate unidirectional traffic flows for a group of protocols, we utilized the distributed internet traffic generator (D-ITG), which can generate TCP, UDP, Telnet, VoIP, DCCP, DNS, ICMP, and SCTP traffic. Third, the OSTINATO tool was used to generate different traffic flows with different configurations for a variety of protocols such as VLANs, network news transfer protocol (NNTP), real-time streaming protocol (RSTP), internet group management protocol (IGMP), and multicast listener discover (MLD) protocol. As shown in Table 2, the total duration of the collected OpenFlow traffic from the physical environment is equal to 137 h with 250 GB total traffic size and the duration of the simulated OpenFlow traffic is 100 h with 143 GB total traffic size.

To generate malicious OpenFlow traffic, we used Hping3 and LOIC (Low Orbit Ion Cannon) in physical and simulated SDN environments. Using these tools, we were able to launch 31 saturation attacks that cover TCP-SARFU, TCP-SYN, ICMP, UDP, IP-Spoofing, and their combinations. As depicted in Table 2, the total size of the physical malicious OpenFlow traffic is 50 GB with a 30 min duration for each attack session, where the simulated malicious traffic has a total size of about 100 GB and the duration of the attack session is 20 min.

**Table 2.** Physical and simulated OpenFlow traffic description.

| Environment Type | Traffic Type | Num. of Session | Duration per Session | Duration | Size |
|---|---|---|---|---|---|
| Physical Environment | Benign Traffic | 104 h | 1–4 h | 137 h | 250 GB |
| | Attack Traffic | 31 | 30 min | 15.5 min | 50 GB |
| Simulated Environment | Benign Traffic | 100 | 1 h | 100 h | 143 GB |
| | Attack Traffic | 31 | 20 min | 10.3 h | 100 GB |

*4.2. Feature Extraction and Data Preprocessing*

The OpenFlow packets have multiple attributes including, but not limited to, message type, packet time, packet length, and source and destination IP addresses. These packets are exchanged between the controller and OpenFlow switches. Formally, OpenFlow traffic (OF) is a collection of OpenFlow packets $<p_1, p_2, \ldots, p_n>$ transmitted and captured in the course of a network (normal or attack) session, where each packet $pi$ has the attributes $<time, srcIP, dstIP, OFmsg, length>$. The OpenFlow protocol v1.3 has 29 OpenFlow message that are generally divided into three categories: (1) controller-to-switch messages, generated by the SDN controller and forwarded to the OpenFlow switch for obtaining information about or changing the state of the switch (e.g., Flow-Mod); (2) asynchronous messages, initiated by the OpenFlow switch without solicitation from a controller and forwarded to the controller to notify about the new received packets and any errors (e.g., Packet-in); and (3) symmetric messages, generated from both sides, such as the Hello message.

From the captured OpenFlow traffic, four features are extracted: the number of Packet-In messages, the number of Packet-Out messages, the number of Flow-Mod messages, and the number of TCP-ACK messages exchanged between the switch to the controller. The extracted features are sensitive to saturation attacks as well as hybrid saturation attacks, which are mainly a combination of different saturation attacks that target the SDN environment. The saturation and hybrid saturation attacks have a different impact on these features. Figure 3 demonstrates the impact of the UDP attack on the presence of the Packet-In and Packet-Out messages on the OpenFlow traffic. At the early stages of the attack, the number of the Packet-In messages increases significantly since the attack packets do not match the flow rules of the OpenFlow switch (i.e., table miss), which leads to generating Packet-In messages that are forwarded to the SDN controller. At the later stages of the attack, the number of Packet-In messages will be dropped followed by an increase in Packet-Out messages generated by the controller as a response to the generated Packet-In message at the early stages of the attack. It is highly important to note here that the SDN controller cannot process every Packet-In message in timely manner and generate a Packet-Out message due to the negative impact of the attack on the SDN environment computational resources. Table 3 explains the impact of TCP-SARFU, UDP, TCP-SYN, ICMP, and IP spoofing attacks on each feature.

Central to detecting saturation attacks is determining the proper time window. Thus, we analyze the impact of various time windows on the prediction results of the K-NN, NB, and SVM classifiers. From the captured OpenFlow traffic, we extracted a separate dataset for each different time-window. This is done in both simulated and physical SDN environments. The time window of the extracted dataset ranges from one minute to the

attack duration. In other words, we extract the datasets from the collected OpenFlow traffic, where the duration of each sample inside the extracted dataset is equal to the defined time window.

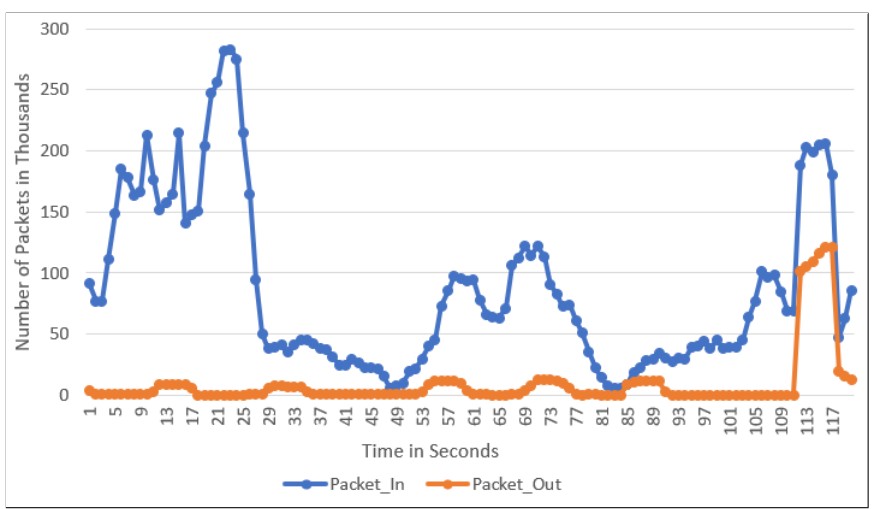

**Figure 3.** The UDP attack impact on the Packet-Out and Packet-In messages.

**Table 3.** The impact of saturation attacks on the main OpenFlow messages.

| Attacks Type | Number of Packet-In | Number of Packet-Out | Number of Flow-Mod | Number of TCP-ACk |
|---|---|---|---|---|
| UDP | ⇑ followed-by ⇓ | ⇓ | ⇓ | ⇑ |
| TCP SYN | ⇑ | ↑ followed-by ⇓ | ↑ followed-by ⇓ | ⇑ |
| ICMP | ⇑ | ↑ followed-by ⇓ | ↑ followed-by ⇓ | ↑ followed-by ⇓ |
| TCP SARFU | ↑ followed-by ⇓ | ↑ followed-by ⇓ | ↑ followed-by ⇓ | ↑ followed-by ⇓ |
| IP Spoofing | ↑ followed-by ⇓ | ↑ followed-by ⇓ | ↑ followed-by ⇓ | ↑ followed-by ↓ |

↑ (Notable-Increase), ⇑ (Significant-Increase), ↓ (Notable-Decrease), and ⇓ (Significant-Decrease).

**Dataset extraction process.** Algorithm 1 demonstrates the process of extracting datasets from the captured OpenFlow traffic. The algorithm requires four inputs as follows: (1) *OF* represents the captured OpenFlow traffic; (2) *T* is the time window of the extracted dataset; (3) *S* is the dataset time shifting; and (4) *L* is the type of the OpenFlow traffic which could be normal or attack traffic. The output is a dataset $X_J = \{x_1, x_2, x_3, ..., x_n\}$, which is a sequence of labeled samples, and each sample $x_j$ contains the following features: number of Packet-In messages, number of Flow-Mod messages, number of Packet-Out messages, number of TCP_ACK messages, and the label which equals to "0" when the sample is extracted from normal traffic and "1" when the sample is extracted from attack traffic. The algorithm extracts the selected features from the packet sequence of OpenFlow traffic *OF* within the defined time window *T* (lines 9–16). Then, for each OpenFlow packet in *OF*, we extract the type of the OpenFlow message. The message type is compared with **our features**, and if match occurs, then the corresponding counter is incremented by one. Afterward, the packet time and the starting time are compared; if the difference between them is larger than time window *T*, the dataset generation tool creates a new sample $x_j$ with the corresponding label, and **the dataset index *J* is incremented** by one (lines 6–8). Line 17 shows how after creating a new sample $x_j$, the algorithm updates the value of *firstPacketIndex*, where the shifting parameter *S* is added to the starting index of the next shift. In other words, we divide the captured OpenFlow traffic into many parts, where the length of each part equals *T*. Then, we count the number of Packet-In, Packet-Out, Flow-Mod, and TCP-ACK messages of each part. The generated dataset has many samples,

where each sample represents each part of length $T$. We use the shifting parameter to simulate different starting points of DDoS attacks and increase the amount of overlap in the extracted dataset.

---

**Algorithm 1:** OpenFlow Dataset Generation

---

**Input:** OpenFlow traffic $OF$, Time-Window $T$, Time-Shifting $S$, Traffic-Type $L$ {0,1}.
**Output:** Dataset $X_J$.
**Declare:** $packetIn, packetOut, packetMod, tcpAck, Msg_{(type)}$

1   $J = 0$ (the index of the output sample $X_J$)
2   **repeat**
3     $firstPacketIndex = 1$    // (Pi is the $1^{st}$ packet of the current shift)
4     $startTime = P_{time}(firstPacketIndex)$ /* (is the $1^{st}$ packet time of the
        current shift)                                                        */
5     **for** *(i = firstPacketIndex; i< n; i++)* **do**
6       **if** $P_{time} - startTime > T$ **then**
7         $createNewSample_{xj\,(packetIn, packetOut, packetMod, tcpAck, L, increment\,J)}$
8         add a new sample $x_j$ to $X_j$ with traffic type $L$ and increase $J$ by one.
9       **if** $Msg_{type}$ = *OFPT PACKET IN* **then**
10         $packetIn+ = 1$
11       **else if** $Msg_{type}$ = *OFPT PACKET OUT* **then**
12         $packetOut+ = 1$
13       **else if** $Msg_{type}$ = *OFPT PACKET MOD* **then**
14         $packetMod+ = 1$
15       **else if** $Msg_{type}$ = *TCP ACK* **then**
16         $tcpAck+ = 1$
17     $firstPacketIndex = updatePacketIndexForNextShift(firstPacketIndex, S)$
18   **until** $firstPacketIndex > n$
19   **return** $X_J$

---

### 4.3. Supervised and Semi-Supervised Machine Learning Classifiers

We utilize the K-nearest neighbor (K-NN), naive Bayes (NB), and support vector machine (SVM) as supervised classifiers and train them using the extracted dataset (see Section 4.2) and evaluate their performance results in detecting known and unknown saturation attacks. In addition to the supervised classifiers, we evaluate the robustness of the semi-supervised machine-learning classifiers in detecting the known and unknown saturation attacks. In our experiments, we utilize the most widely used semi-supervised classifiers, such as basic autoencoder, isolation forest, one-class SVM, and variational autoencoder.

### 4.4. The Hybrid Approach

Using a single supervised classifier to detect saturation attacks leaves SDNs vulnerable to unknown saturation attacks [9]. The supervised classifiers require a large size of training dataset that includes large samples of each type of attack. This is because the supervised classifiers cannot generalize enough to detect any unknown attack. Therefore, the SDN environment will be vulnerable to flooding attacks.

In this paper, we study the effectiveness of combining supervised and semi-supervised classifiers to detect known and unknown attacks against SDNs. The architecture of the hybrid approach comprises two classification algorithms: First, a supervised classifier to detect the known attacks. In our case, the K-NN classifier is adopted since it provides the highest detection performance for known attacks among the other supervised classifiers evaluated in this work. Second, we adopt the variational autoencoder as a semi-supervised classifier to detect unknown attacks due to it is high performance in detecting saturation attacks, as reported in the experimental results.

The hybrid approach is a two-stage process: the training stage and the detection stage. In the training stage, a pre-made datasets should be used to train the two classifiers. The training dataset of the supervised classifiers should include labeled observations for known attacks and normal traffic. In this work, the training dataset of the semi-supervised classifier includes observations of normal OpenFlow traffic. In the detection stage, if any of the classifiers or both of them raise an alert, an attack is detected. Our offline experiments show that the hybrid approach provides a higher detection performance for known and unknown attacks than using a single supervised or semi-supervised classifier.

## 5. Experiment Results and Discussions

The experiment focuses on the following research questions:

**RQ1:** Is it possible to find the proper time window for performing OpenFlow traffic analysis, and if so, what is it?

**RQ2:** How is the detection performance of a classifier affected by the variation of the time window?

**RQ3:** Can the supervised and semi-supervised classifiers detect the unknown saturation attacks?

**RQ4:** Are semi-supervised classifiers robust to noise during the training phase?

**RQ5:** Does the hybrid approach (see Section 4.4) reap the benefits of both semi-supervised and supervised classifiers in detecting known and unknown saturation attacks?

### 5.1. The Proper Time Window for Detecting Known Attacks

For the physical environment, 30 datasets were extracted using the collected OpenFlow traffic. Each dataset corresponds to a different time window of OpenFlow traffic analysis. The time windows range from 1 min to 30 min. Each dataset was used to train and test K-NN, NB, and SVM classifiers. The recall, precision, and F-1 score were employed to evaluate the detection performance of our trained models and analyze the impact of different time windows on the prediction results of our models.

Figure 4a demonstrates the precision, recall, and F1 score of the trained K-NN models using the datasets of different time windows that range from 1 min to 30 min. The K-NN classifier achieved the highest detection performance rate when the time window was equal to one minute of the OpenFlow traffic analysis: the recall was 95% with precision 96% and 95% F1 score. The lowest detection performance rate was obtained when the time window was equal to 30 min, where the corresponding recall is 98% with precision 47% and 64% F1 score. Thus, in the physical environment, the proper time window of the K-NN classifier is equal to one minute. To support this conclusion, Figure 4a demonstrates that the precision and F-1 score rates declined as the time window of the OpenFlow traffic increased. Figure 4b presents the detection performance results for the trained SVM models.

The SVM classifier obtains the highest detection results when the time window equals one minute, in which the recall of 92%, precision is 91%, and the F1 score is 91%. When the time window equals 30 min, the SVM trained model obtains the lowest detection performance rate with a recall of 99%, precision 46%, and an F1 score of 62%. Figure 4b shows the impact of the time-window length on the SVM classifier detection performance. Particularly, the precision and F1 score values are decreased, and the recall value is increased as the time window is increased. Therefore, in the physical environment, the SVM classifier's proper time window to detect saturation attacks equals one minute.

Similarly, we evaluated the NB classifier detection performance by conducting 30 experiments. For each experiment, different training and testing datasets of different time windows were used. As shown in Figure 4c, the NB classifier obtains the highest detection performance when the time window is equal to three minutes with 80% recall, 99% precision, and an F1 score of 89%. The lowest detection results are obtained by the NB trained model when the time window equals 30 min with 53% recall, 52% precision, and an F1 score of 52%.

By using the OpenFlow traffic simulation environment, 20 datasets were extracted. Each dataset represents a different time window of OpenFlow traffic analysis. The datasets' time window ranges from 1 min to 20 min. These datasets were used to evaluate the classifiers. In each experiment, we recorded the recall, precision, and F-1 score for the trained models to evaluate their detection performance.

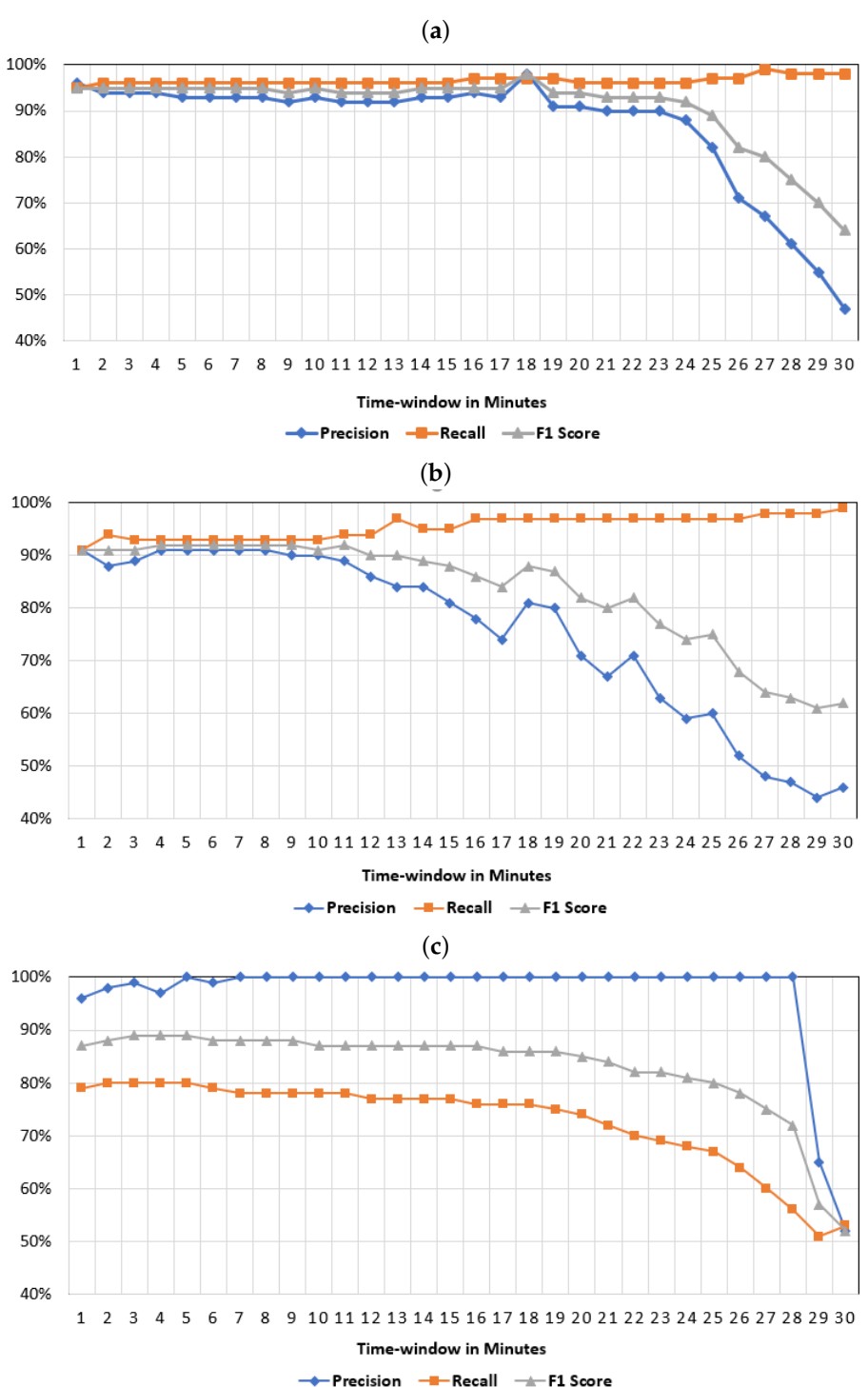

**Figure 4.** The detection performance of supervised classifiers using SDN physical environment. (**a**) K-NN performance on different time windows on physical environment; (**b**) SVM detection performance on different time windows on physical environment; (**c**) NB detection performance on different time windows on physical environment.

As shown in Figure 5a, the K-NN classifier achieves the highest detection performance when the time window equals one minute with 99% recall, 97% precision, and 98% F-1 score. Furthermore, the K-NN classifier achieves the highest precision rate (100%) when the time window is equal to 18 min, but with very low recall (19%) and F1 score (35%). Thus, the one-minute time window is a proper choice for the K-NN classifier to detect the saturation attacks with detection of the highest performance in the simulated SDN environment. Additionally, Figure 5a demonstrates the impact of increasing the time window of OpenFlow traffic analysis on the detection results of K-NN trained models; a longer time window decreases the K-NN classifiers detection models performance. For the SVM classifier, Figure 5b presents the detection performance using different time windows. The SVM trained model acquires the highest detection rate with 89% recall, 81% precision, and 85% F1 score when the time window is equal to 2 min. When the time window is equal to 20 min, the detection performance is decreased drastically to 35% recall, 7% precision, and 11% F1-score.

The NB classifiers trained models did not behave differently from K-NN and SVM. When the time window increases, the detection performance decreases and vice versa. As shown in Figure 5c, the NB classifier has the highest detection performance with 96% recall, 85% precision, and 91% F1 score when the time window is equal to one minute. When the time window is equal to 20 min, the NB classifier has the lowest detection performance with 37% recall, 11% precision, and 17% F1 score. The experimental results in both the physical and simulated SDN environments demonstrate the crucial rule of time window on the detection performance of machine learning classifiers. A longer time window led to a decrease in the detection performance of saturation attacks.

## 5.2. The Impact of Time Window Variation on the Detection Performance of Machine Learning Classifiers

Based on the results of the experiments in Section 5, the time window has a significant impact on the detection performance of our classifiers. A longer time window led to a decrease in the overall results of our evaluation metrics and detection performance. Figure 4a shows the metric values from the K-NN classifier applied to the physical environment. The precision ratios decrease as the time window increases. The recall ratios increase slightly and the F1 scores decrease as the time window is increased, but the overall detection performance of the K-NN classifier decreases. Similarly, the SVM classifier detection results in the physical and simulated experiments showing a similar behavior to the K-NN classifier. Figures 4b and 5b show the evaluation results of the SVM experiments on the physical and simulated environments, respectively. The precision rates, recall rates, and F1 score rates decrease when the time window increases. The same behavior is confirmed by NB trained models, in which their detection performance declines when the length of the time window is increased (see Figures 4c and 5c).

To further investigate the role of the time window length on the detection results of the supervised classifiers, we perform the false negative analysis. The main objective of this analysis is to allocate all the misclassified samples by the trained classifiers as well as the corresponding time slot of these samples. During the false-negative analysis, we extract the attacks samples and the associated time slot for each sample from the testing datasets. Then, we fed all the attack samples to the trained models of K-NN, SVM, and NB for prediction. The results of the false-negative analysis revealed that the majority of the false negatives happened at the end of the saturation attack. As depicted in Figure 4, the precision, recall, and F-1 score of the trained models decreased dramatically when the time window was equal to 15 min or longer. This is due to the status of the SDN environment. Basically, at the initial phase of the attack, the controller and the OpenFlow switches have enough capacity to process the table miss packets. Furthermore, the OpenFlow channel has enough bandwidth to transfer OpenFlow messages, which led to a huge increase in the numbers of Packet-In, Flow-Mod, Packet-Out, and TCP-ACK messages. Therefore, the trained classifiers can identify the attack samples with high recall, precision, and F-1 score.

Later, when the attack takes-over the SDN environment, the computational resources of the controller and the OpenFlow switch were consumed. In this case, they do not have the required resources to process malicious traffic. Thus, the occurrences of Packet-In, Packet-Out, Flow-Mod, and TCP-ACK messages in the malicious traffic will be similar to their occurrences in the normal traffic. As a result, the false negatives are increased since the trained classifiers classify the attack samples as normal ones, which reduces the recall ratio or increases the false positives that decrease the precision ratio of the trained classifiers.

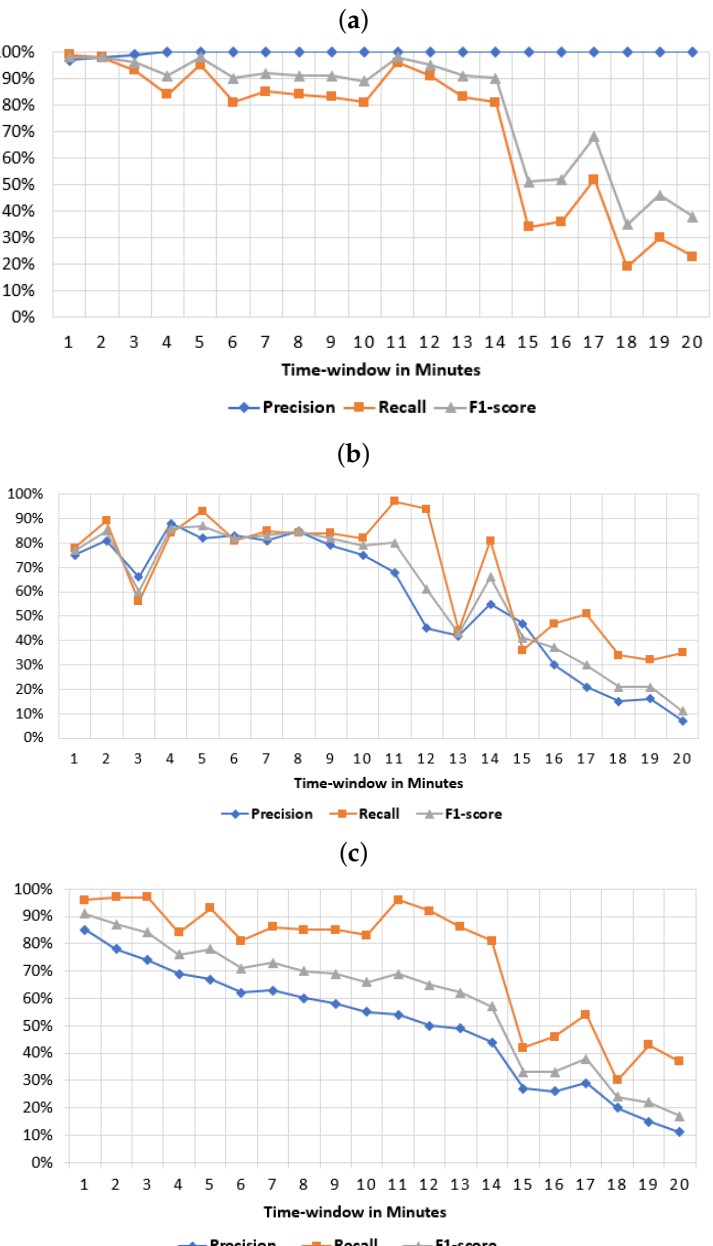

**Figure 5.** The detection performance of supervised classifiers using SDN simulation environment. (**a**) K-NN performance on different time windows on simulation environment; (**b**) SVM detection performance on different time windows on simulation environment; (**c**) NB detection performance on different time-windows on simulation environment.

### 5.3. Detecting Unknown Saturation Attacks Using Semi-Supervised Machine Learning

It is challenging for machine-learning-based detection systems to detect new attacks which the classifier has not yet observed in the training phase. This is because the standard

approach of the ML-based detection system is to train the classifier to detect all known attack classes. Therefore, the training datasets should contain a large number of specimens for each attack class and include all real-life attacks [25], which is hard to obtain. Thus, ML-based DDoS detection systems are not widely used in real-life environments because they cannot detect the unknown attacks. In this paper, we investigate the detection performance of SVM, K-NN, and NB supervised classifiers for identifying unknown saturation attacks. An unknown attack is an attack that has been mislabeled by the trained model due to the absence of similar samples in the training dataset. Then, we compare the detection performance of the semi-supervised classifiers in detecting unknown attacks. In our approach, the semi-supervised classifiers are trained using the normal datasets that consists of only normal traffic samples.

In our experiments, in order to create unknown attacks, the targeted attack observations and its combinations are excluded from the training dataset. Hence, the training dataset includes only samples of other attacks, their combinations, and normal traffic samples. The testing dataset is composed of the unknown attack samples and randomly selected normal traffic samples. Assuming that $R$ is the training dataset which includes all attack samples and normal traffic samples, $W$ is the testing dataset that includes both normal and attack samples, $U$ is the unknown attack observations, and $Z$ is the observations of attack combination, then

$$R_{trainingDataset} = R - U - Z \tag{1}$$

$$W_{testingDataset} = U + NormalTrafficSamples \tag{2}$$

Using the best-case training method in which one type of attack is excluded from the training dataset, the K-NN, SVM, and NB classifiers can detect the unknown attack as depicted in Figure 6. Specifically, the K-NN classifier obtains high detection results in detecting unknown saturation attacks. We hypothesize that all the different attacks in the training set allow the classifier to generalize the missing attack. To confirm this hypothesis, we use the worst-case training method. In this case, the training dataset has only one type of attack in addition to the normal samples. In our experiments, we consider the K-NN classifier (our best choice) only, and it is trained using one type of attack. The results in Table 4 show that the K-NN, in the worst-case scenario, is not able to efficiently detect the unknown attacks. This means, in general, that supervised classification is unsuitable for detecting unknown SDN attacks. For these reasons, we consider studying semi-supervised classifiers, such as isolation forest, one-class SVM, autoencoder, and variational autoencoders for attack detection.

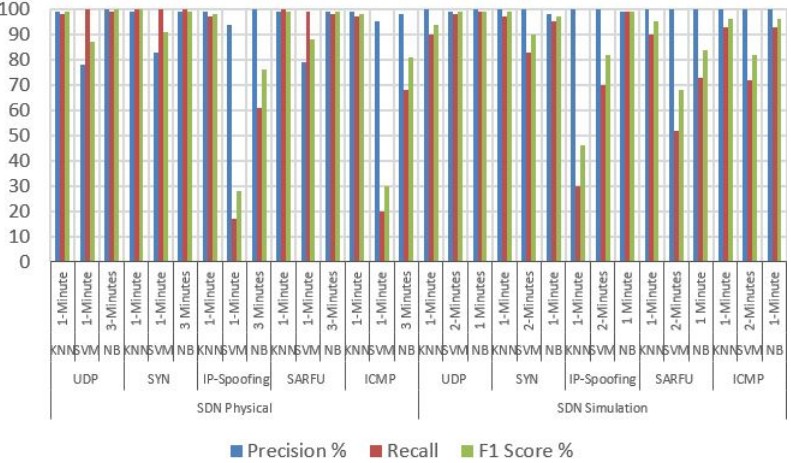

**Figure 6.** The performance of supervised classifiers in detecting unknown saturation attacks using the best-case training method.

In this case, the training set comprises only the normal instances, whereas the testing set consists of all the normal and attack instances. Table 5 shows the average results of the 10-fold cross validation. In the worst-case scenario, the semi-supervised classification algorithms have higher detection performance of unknown saturation attacks with respect to supervised ones. Specifically, the variational autoencoder is effective in the detection of unknown saturation attacks and obtains comparable results with respect to the supervised classifiers in detecting known saturation attacks.

Nonetheless, we demonstrated that the variational autoencoder classifier can detect unknown saturation attacks with relatively high accuracy. This is due to different reasons. First, the traffic of flooding attacks has a high degree of similarity [2]. Meanwhile, the normal OpenFlow traffic has a low degree of self-similarity and has different statistical characteristics. Second, the selected features are sensitive to any abnormal behavior of the OpenFlow traffic since they represent the key messages of the OpenFlow protocol.

**Table 4.** K-NN 1 min trained with only one attack and tested on all the attacks.

| | Physical Environment | | | Simulated Environment | | |
|---|---|---|---|---|---|---|
| **Attack** | **Precision** | **Recall** | **F1-Score** | **Precision** | **Recall** | **F1-Score** |
| UDP | 100% | 22% | 55% | 100% | 84% | 85% |
| TCP-SYN | 100% | 2% | 4% | 100% | 80% | 89% |
| TCP-SARFU | 100% | 23% | 37% | 100% | 82% | 90% |
| ICMP | 100% | 16% | 28% | 95% | 83% | 91% |
| IP-Spoofing | 99% | 11% | 20% | 97% | 61% | 75% |

**Table 5.** Semi-supervised classification algorithm tested on all the attacks (1 min).

| | Physical Environment | | | Simulated Environment | | |
|---|---|---|---|---|---|---|
| **Algorithm** | **Precision** | **Recall** | **F1-Score** | **Precision** | **Recall** | **F1-Score** |
| Isolation Forest | 56% | 92% | 69% | 38% | 100% | 56% |
| One-Class SVM | 22% | 97% | 35% | 11% | 100% | 20% |
| Basic Autoencoder | 99% | 24% | 38% | 96% | 80% | 86% |
| variational Autoencoder | 86% | 93% | 90% | 85% | 97% | 91% |

*5.4. Robustness of Semi-Supervised Machine Learning*

This section investigates the variational autoencoder robustness against the noise of the training dataset. The variational autoencoder assumes that all the training examples belong to a unique class, in our case, benign examples. However, during the initial setup of the saturation attack detection systems, some machines are connected to the SDN network, which may have abnormal behavior or, during data collection, some attacks are already perpetrated. Therefore, we conducted multiple experiments using the variational autoencoder with a training dataset that includes a small percentage of attack observation (i.e., noisy samples), which are very close to the benign ones. The attack examples are selected by using the k-nearest neighbor data structure, which was trained using benign examples. Figures 7 and 8 show the detection performance results of the variational autoencoder in terms of precision, recall, and F1-score with different noise percentages, for both physical and simulated SDN environments. We can observe that the variational autoencoder is robust and maintains comparable high performance, even with a high noise ratio. We also observe that the physical experiments demonstrate better robustness than the simulated experiment. In fact, with 15% noise, the F1-score and recall are smaller than 0.8, while in the simulated experiments they are always above 0.8. Therefore, we chose to utilize the variational autoencoder in our hybrid approach.

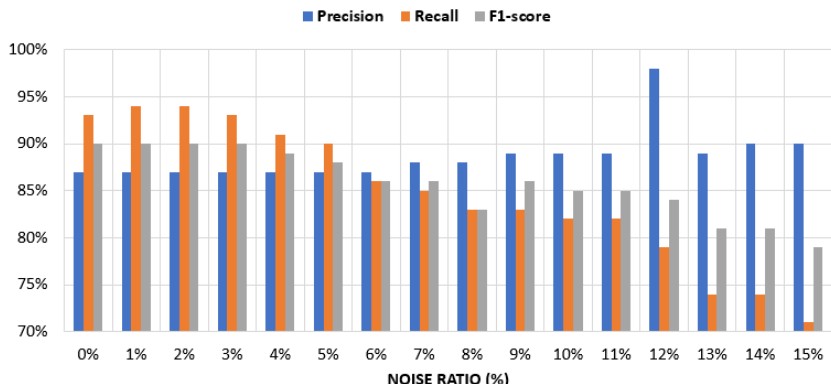

**Figure 7.** Variational autoencoder robustness experiment results on physical environment.

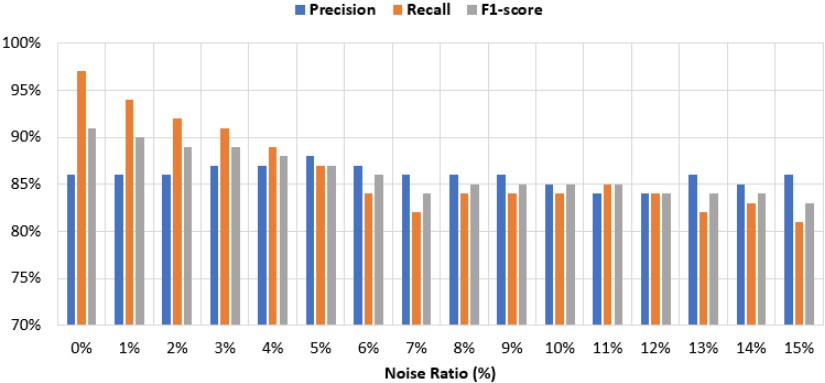

**Figure 8.** Variational autoencoder robustness experiment results on simulation environment.

### 5.5. The Detection Performance of the Hybrid Approach

In this section, we present the detection performance of the hybrid approach and compare its results with K-NN and variational autoencoder. In our offline case study, we conducted 31 experiments that includes $SYN, SARFU, IP\text{-}Spoofing, ICMP, and SARFU$ attacks and their combinations. During our experiments, we divide the attacks into different sets. For instance, given each set, we assume that its attacks are known to the trained model and the remaining attacks are unknown. For example, considering $W$ as a set consists of {IP-Spoofing, SARFU, ICMP} as known attacks, whereas UDP and SYN attacks are considered unknown attacks.

In each experiment, we report the average precision, recall, and F1-score among all 10-fold supervised approaches (K-NN), variational autoencoder, and the hybrid approach. Table 6 reports the results of the experiments in the physical and simulated environments. We observe that the hybrid approach always significantly outperforms the K-NN approach in terms of recall and F1-score. In terms of precision, the K-NN result is higher. However, it is important to note that it achieves higher precision due to its poor recall, i.e., it detects a smaller number of attacks, but more precisely. In comparison with the variational autoencoder, the hybrid approach achieves a similar or improved performance in terms of precision, recall, and F1-score.

**Table 6.** Hybrid approach vs. K-NN and variational autoencoder trained with a specific set of known attacks.

| | Physical Environment | | | | | | Simulated Environment | | | | | |
|---|---|---|---|---|---|---|---|---|---|---|---|---|
| | Precision | | Recall | | F1-Score | | Precision | | Recall | | F1-Score | |
| **Variational Autoencoder** | 0.86 | | 0.93 | | 0.90 | | 0.85 | | 0.97 | | 0.91 | |
| **Known Attacks** | **K-NN** | **Hybrid** | **K-NN** | **Hybrid** | **K-NN** | **Hybrid** | **K-NN** | **Hybrid** | **K-NN** | **Hybrid** | **K-NN** | **Hybrid** |
| {ICMP} | 1.00 | 0.86 | 0.16 | 0.94 | 0.28 | 0.90 | 1.00 | 0.90 | 0.83 | 0.99 | 0.91 | 0.94 |
| {IP-Spoofing} | 0.99 | 0.87 | 0.11 | 0.97 | 0.20 | 0.91 | 0.97 | 0.90 | 0.61 | 0.98 | 0.75 | 0.94 |
| {IP-Spoofing,ICMP} | 0.99 | 0.87 | 0.20 | 0.96 | 0.33 | 0.91 | 0.98 | 0.90 | 0.92 | 1.00 | 0.95 | 0.95 |
| {IP-Spoofing,SARFU} | 1.00 | 0.87 | 0.35 | 0.96 | 0.52 | 0.91 | 0.99 | 0.90 | 0.88 | 0.99 | 0.93 | 0.94 |
| {IP-Spoofing,SARFU,ICMP} | 1.00 | 0.87 | 0.43 | 0.96 | 0.60 | 0.91 | 0.99 | 0.91 | 0.90 | 0.98 | 0.94 | 0.94 |
| {SARFU} | 1.00 | 0.87 | 0.23 | 0.94 | 0.37 | 0.90 | 1.00 | 0.90 | 0.82 | 0.97 | 0.90 | 0.94 |
| {SARFU,ICMP} | 1.00 | 0.86 | 0.40 | 0.94 | 0.57 | 0.90 | 1.00 | 0.91 | 0.86 | 0.99 | 0.93 | 0.94 |
| {SYN} | 1.00 | 0.86 | 0.02 | 0.93 | 0.04 | 0.90 | 1.00 | 0.90 | 0.80 | 0.99 | 0.89 | 0.94 |
| {SYN,ICMP} | 1.00 | 0.87 | 0.31 | 0.93 | 0.47 | 0.90 | 1.00 | 0.90 | 0.84 | 0.98 | 0.91 | 0.94 |
| {SYN,IP-Spoofing} | 0.99 | 0.87 | 0.30 | 0.97 | 0.47 | 0.92 | 0.98 | 0.90 | 0.90 | 1.00 | 0.94 | 0.95 |
| {SYN,IP-Spoofing,ICMP} | 0.99 | 0.87 | 0.38 | 0.97 | 0.55 | 0.92 | 0.98 | 0.90 | 0.93 | 1.00 | 0.95 | 0.95 |
| {SYN,SARFU} | 1.00 | 0.86 | 0.25 | 0.94 | 0.41 | 0.90 | 1.00 | 0.90 | 0.85 | 1.00 | 0.92 | 0.95 |
| {SYN,SARFU,ICMP} | 1.00 | 0.86 | 0.54 | 0.93 | 0.70 | 0.90 | 1.00 | 0.90 | 0.87 | 0.97 | 0.93 | 0.93 |
| {SYN,SARFU,IP-Spoofing} | 1.00 | 0.87 | 0.54 | 0.97 | 0.70 | 0.91 | 0.99 | 0.90 | 0.89 | 1.00 | 0.93 | 0.95 |
| {SYN,SARFU,IP-Spoofing,ICMP} | 1.00 | 0.87 | 0.90 | 0.98 | 0.90 | 0.92 | 0.98 | 0.90 | 0.95 | 1.00 | 0.96 | 0.95 |
| {UDP} | 1.00 | 0.86 | 0.22 | 0.93 | 0.36 | 0.90 | 1.00 | 0.90 | 0.74 | 0.97 | 0.85 | 0.94 |
| {UDP,ICMP} | 1.00 | 0.86 | 0.38 | 0.94 | 0.55 | 0.90 | 1.00 | 0.90 | 0.84 | 0.97 | 0.91 | 0.94 |
| {UDP,ICMP,IP-Spoofing,SYN} | 1.00 | 0.87 | 0.71 | 0.97 | 0.83 | 0.92 | 0.99 | 0.90 | 0.91 | 0.99 | 0.93 | 0.94 |
| {UDP,ICMP,SYN} | 1.00 | 0.86 | 0.64 | 0.94 | 0.78 | 0.90 | 1.00 | 0.90 | 0.85 | 0.98 | 0.92 | 0.94 |
| {UDP,IP-Spoofing} | 1.00 | 0.87 | 0.33 | 0.96 | 0.50 | 0.91 | 0.99 | 0.90 | 0.87 | 0.98 | 0.92 | 0.94 |
| {UDP,IP-Spoofing,ICMP} | 1.00 | 0.87 | 0.42 | 0.96 | 0.59 | 0.91 | 0.99 | 0.90 | 0.87 | 0.99 | 0.93 | 0.94 |
| {UDP,IP-Spoofing,SYN} | 1.00 | 0.87 | 0.65 | 0.97 | 0.78 | 0.92 | 1.00 | 0.90 | 0.85 | 0.98 | 0.92 | 0.94 |
| {UDP,SARFU} | 1.00 | 0.86 | 0.24 | 0.93 | 0.39 | 0.90 | 1.00 | 0.90 | 0.86 | 0.99 | 0.93 | 0.94 |
| {UDP,SARFU,ICMP} | 1.00 | 0.87 | 0.41 | 0.94 | 0.58 | 0.90 | 1.00 | 0.90 | 0.87 | 0.97 | 0.93 | 0.93 |
| {UDP,SARFU,ICMP,SYN} | 1.00 | 0.86 | 0.89 | 0.93 | 0.90 | 0.90 | 1.00 | 0.90 | 0.88 | 0.98 | 0.94 | 0.94 |
| {UDP,SARFU,IP-Spoofing} | 1.00 | 0.87 | 0.36 | 0.96 | 0.53 | 0.91 | 0.99 | 0.90 | 0.91 | 0.99 | 0.94 | 0.94 |
| {UDP,SARFU,IP-Spoofing,ICMP} | 1.00 | 0.87 | 0.92 | 0.96 | 0.91 | 0.91 | 1.00 | 0.90 | 0.90 | 1.00 | 0.93 | 0.95 |
| {UDP,SARFU,IP-Spoofing,SYN} | 1.00 | 0.87 | 0.66 | 0.97 | 0.80 | 0.92 | 1.00 | 0.90 | 0.87 | 0.98 | 0.93 | 0.94 |
| {UDP,SARFU,SYN} | 1.00 | 0.86 | 0.39 | 0.94 | 0.56 | 0.90 | 1.00 | 0.91 | 0.87 | 0.97 | 0.93 | 0.93 |
| {UDP,SYN} | 1.00 | 0.87 | 0.37 | 0.94 | 0.54 | 0.90 | 1.00 | 0.90 | 0.82 | 0.99 | 0.90 | 0.94 |

## 6. Threats to Validity

The results of this work heavily rely on the experiments conducted in both physical and simulated SDN environments. In this section, we discuss various factors that may affect the validity of the results. The discussion not only clarifies the strengths and assumptions of this work, but also facilitates future research on the detection of saturation attacks in SDN (e.g., replicating the experiments and comparing a new approach to this work).

**Threats to external validity** are concerned about generalizing our findings. The results of our experiments show the differences between the physical and the simulated environments in terms of the detection performance of the K-NN, SVM, and NB classifiers and the proper time window to detect known and unknown saturation attacks. For instance, the proper time window for the NB classifier to detect the known and unknown saturation attacks in the physical environment was 3 min. In the simulation environment, the proper time window was 1 min of OpenFlow traffic analysis. This raises a concern related to the relationship between the setup of the SDN environment and the task of allocating the proper time window for the detection of saturation attacks. In our study, the physical environment is a relatively small SDN environment, which consists of an OpenFlow vSwitch and five hosts. We were able to replicate real-world internet traffic using different traffic generation tools. In contrast with the simulated environment, we were able to configure medium-to-large scale SDN environments with some limitations of mimicking real-world network traffics. Thus, we believe that the scale of the SDN network and the network traffic are important parameters that influence identifying the proper time window for each machine-learning classifier. Therefore, we cannot guarantee that the proper time windows of OpenFlow traffic analysis that were obtained in this research, by their corresponding machine learning classifiers, are applicable to all SDN environments. Specifically, different SDN environments might have different proper time windows of OpenFlow traffic analysis for the same task (detection of saturation attacks). However, our analysis provides the basis for further research to obtain the proper time window for different SDN environments.

**Threats to construct validity** concern the relationship between the construct and observation. One issue is related to the dataset mislabeling noise and the proper time window for detecting saturation attacks. Our experiments resulted in a small number of normal samples that were mislabeled as attack samples in the generated datasets. Also relevant is the relationship between the dataset mislabeling noise and the identification of the proper time window. The mislabeling noise of training datasets may influence the detection performance of the classifiers. The choice of the proper time window of OpenFlow traffic analysis for the classifiers may change with different datasets. However, in this research, we used a large number of correctly sampled normal traffic in order to minimize the impact of measurement noise coming from a few mislabeled training samples. In addition, we evaluated the robustness of the variational autoencoder classifier against the noise of training datasets, but not all the supervised machine learning classifiers were evaluated against the training dataset noisiness.

**Threats to internal validity** concern the internal parameters that could influence the variables and the relations being studied. In our experiments, we used the suggested configurations for SVM and NB classifiers, for instance, the SVM kernel type. However, some parameters could be configured that could affect the obtained proper time window of the OpenFlow traffic analysis.

**Threats to conclusion validity** concern the treatment and the findings. The generated datasets have attack samples that are extracted from the malicious OpenFlow traffic. It comprises the saturation attacks that are mostly used by hackers to compromise the SDN networks. While we considered state-of-the-art saturation attacks, we cannot guarantee that our datasets covered all saturation attacks in SDN. Additionally, we did not investigate all the supervised machine learning classifiers in detecting the unknown saturation attacks. Thus, some of these classifiers could have the capability of detecting these kinds of attacks.

## 7. Conclusions

In this paper, we introduced a comprehensive study that focuses on the efficiency of using supervised and semi-supervised classifiers in detecting the known and unknown DDoS saturation attacks in SDN. We conducted extensive experiments using three supervised classifiers and four semi-supervised classifiers trained on different datasets extracted from real-life and simulated OpenFlow traffic. The reported results demonstrate that the supervised classifiers are effective in detecting the known DDoS attacks and provide higher detection performance than the semi-supervised classifiers. Additionally, the reported results demonstrate that the semi-supervised classifiers can effectively detect unknown DDoS attacks and provide higher detection results than the supervised classifiers. Furthermore, we enhance the detection performance by proposing a hybrid approach that combines the supervised and semi-supervised classifiers to provide reliable detection for both known and unknown DDoS saturation attacks. The results demonstrate that the hybrid approach outperforms the individual supervised and semi-supervised classifiers.

The current work was conducted in SDN networks with a single controller. We believe that the proposed approaches are applicable to multi-controller SDN networks. Our future work will investigate how well the supervised and unsupervised classifiers perform in detecting known and unknown attacks in large-scale SDN environments, where there are a number of SDN controllers.

**Author Contributions:** Data curation, M.A.; Investigation, S.K.; Methodology, S.K. and A.A.-A.; Supervision, S.K.; Writing—original draft, S.K.; Writing—review & editing, A.A.-A. and H.W.A. All authors have read and agreed to the published version of the manuscript.

**Funding:** This research received no external funding.

**Data Availability Statement:** Not Applicable, the study does not report any data.

**Conflicts of Interest:** The authors declare no conflict of interest.

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
