# Peer review of "The Robustness of Detecting Known and Unknown DDoS Saturation Attacks in SDN via the Integration of Supervised and Semi-Supervised Classifiers"

_futureinternet, doi:10.3390/fi14060164_

Round 1

Reviewer 1 Report

This paper seems to be a report for examination of performance of several method of protecting unknown DDOS. It doesn't look for article.

Based on the you examinarion, they proposed just a combined two well known methods. That is, there is no novelty.

Minor problems

  1. The third contribution you argued is not a contribution,
  2. What is “TNP” at line 148 ?
  3. “is” is missed at the line 193. Check all typo-mistake.
  4. Is the second row at the table 2 not “Simulated Environment” ?

The authors argued that only a supervised learning model can detect known DDOS but fails to detect unknown attack and a semi-supervised model can detect well unknown  DDOS attack but  fails to do existent attack. They showed that argument is true with multiple experiments. So, they proposed a hybrid detection model with the combination of a supervised model and an semi-supervised model. 

Practically, it is a very interesting topic, however, theoretically, it is not a novel idea but just a combination of the existing methods.

It is not any added or modified idea with existent methods.

The structure of the paper is good and is written well.

The paper is easy to read 

They showed many results of the many experiments. So, I said it looks like a report, not a paper.

They concluded that their idea is well with an experiment. However, The result is not much superior to the existing experiment.

Reviewer 2 Report

  1. There is a main problem in the introduction section so, the authors should be followed these steps:

It is useful to analyze the issues to be considered in the ‘Introduction’ section under 3 headings. Firstly, information should be provided about the general topic of the article in the light of the current literature which paves the way for the disclosure of the objective of the manuscript. Then the specific subject matter, and the issue to be focused on should be dealt with, the problem should be brought forth, and fundamental references related to the topic should be discussed. Finally, our recommendations for a solution should be described, in other words, our aim should be communicated. When these steps are followed in that order, the reader can track the problem and its solution from his/her own perspective in the light of current literature.

  1. The authors should be shown more results in the abstract.
  2. Revise the title to make it more meaningful.
  3. Explain the novelty of your work presented in this work.
  4. Algorithm 1 is not clear. The authors should clarify more information in it.
  5. Figures 2, and 3 are very important, the authors should clarify more information in them.
  6. Feature Extraction section is not clear. The authors should clarify more information in it.
  7. Conclusion was displayed poorly. the author should review this section.
  8. The authors should add more recently references

Reviewer 3 Report

In this article, Authors present a method to detect known and unkown DDoS attacks over internet. Many AI methods are tested and compared and results are well presented with graphs and tables.

The overall reading of the article is smooth and all the concepts and research questions are well described and supported by results. I do not find any relevant complain about the work.

Author Response

We would like to thank the reviewers for their helpful and insightful comments on our submission. We have attempted to best address each within the revised version of the manuscript. Below in red are our responses and short summaries of how we addressed each comment and the location in the paper (if not obvious) of the update. We have also made the text red of all our changes in our revision to help explicate updates.

No comment has required any response to the reviewer-3

Round 2

Reviewer 1 Report

1. Some paragraphs of the same meaning are duplicated for you to emphasze that supersized method only can't detect unknown DDOS attack. Remove unnecessary paragraphs in your manuscript.

2. In method section(corresponding to proposed  method section), generally, the idea (abstract) is proposed with conceptuall method, without physical (fixed) value like "200GB" . In your method, your method is based on the fixed value. In this case, if the environments are changed, your idea is useless. I recommend that you separate the idea from the physical environments.

3. Also, your idea (hybrid) is proposed after you examine the result from the experiment(section 5). You don't need to describe this. Just propose yor method. In experiment, you show the result of your method which works well for the experimental data.

Reviewer 2 Report

.

Author Response

Thank you very much for your valuable comments that led to improving the quality of the paper.